# Mutational Landmarks in Anaplastic Thyroid Cancer: A Perspective of a New Treatment Strategy

**DOI:** 10.3390/jcm14092898

**Published:** 2025-04-23

**Authors:** Janice Pakkianathan, Celina R. Yamauchi, Luiza Barseghyan, Joseph Cruz, Alfred A. Simental, Salma Khan

**Affiliations:** 1Department of Biochemistry, School of Medicine, Loma Linda University, Loma Linda, CA 92350, USA; jpakkianathan@students.llu.edu (J.P.); ryamauchi@llu.edu (C.R.Y.); lbarseghyan@students.llu.edu (L.B.); josephrcruz035@gmail.com (J.C.); 2Center for Health Disparities & Molecular Medicine, School of Medicine, Loma Linda University, Loma Linda, CA 92350, USA; 3Department of Otolaryngology, School of Medicine, Loma Linda University, Loma Linda, CA 92350, USA; asimenta@llu.edu

**Keywords:** anaplastic thyroid cancer, BRAF-V600E, MEK, p53, TERT, targeted therapy

## Abstract

Anaplastic thyroid carcinoma (ATC) is the rarest and most aggressive form of thyroid cancer, marked by a poor prognosis and resistance to conventional treatments. Like many malignancies, ATC has a complex genetic landscape, with numerous mutations driving tumor initiation, progression, and therapeutic resistance. However, recent advances in molecular research have expanded our understanding of these genetic alterations, paving the way for new targeted treatment strategies. Currently, therapies targeting specific genetic mutations, such as BRAF and MEK, show promise, but their effectiveness is limited to patients harboring these mutations. To explore broader therapeutic possibilities, we conducted a comprehensive literature review using the PubMed database and Google to identify studies on key genetic mutations in ATC. By leveraging these molecular insights, we aim to highlight potential therapeutic avenues that could enhance treatment options and improve patient outcomes.

## 1. Introduction

Anaplastic thyroid carcinoma (ATC) is a highly aggressive and fatal malignancy, accounting for only 2–3% of all thyroid cancer cases [1]. Despite its rarity, ATC is associated with a grim prognosis, with an average survival of just three months, and fewer than 10% of patients surviving beyond one year after diagnosis [2]. The disease is marked by rapid progression and resistance to treatment, posing significant therapeutic challenges [3,4].

Differentiated thyroid cancers (DTCs), such as papillary (PTC) and follicular (FTC) thyroid cancer, respond well to standard radioactive iodine (RAI) therapy, which relies on the sodium/iodide symporter (NIS) for iodine uptake [5,6]. However, ATC’s undifferentiated nature leads to altered NIS expression, rendering it resistant to RAI therapy and necessitating alternative treatment strategies. While efforts are underway to restore NIS expression in ATC, no clinical data currently support its efficacy [5].

ATC is also considered a dedifferentiated cancer, meaning it can arise from previously well-differentiated thyroid cells that have lost their normal structural and functional characteristics [7]. This process of dedifferentiation leads to a highly aggressive and undifferentiated tumor phenotype, attributing to the loss of iodine uptake and resistance to conventional thyroid cancer therapies.

ATC is often associated with differentiated thyroid cancers [5]. Approximately 20% of ATC patients have a documented history of DTC, suggesting that ATC can evolve from pre-existing differentiated malignancies. Moreover, histological analysis reveals that up to 70% of ATC cases have a coexisting DTC component [8].

Molecularly, ATC shares many genetic alterations with DTCs. Mutations in genes commonly found in DTC, such as *BRAF, RAS, TERT promoter*, and *TP53*, are also frequently observed in ATC [9]. One study by Xu et al. reported 58% of ATC cases (n = 360) with concurrent DTC and MAPK pathway mutations also found in DTC [9]. This supports the hypothesis that ATC arises through progressive genetic and epigenetic alterations in pre-existing DTCs, leading to loss of differentiation and increased tumor aggression.

ATC exhibits additional molecular alterations that drive its dedifferentiation and aggressive behavior, including mutations in *PIK3CA*, *PTEN*, and *AKT1*, as well as the inactivation of tumor suppressor genes such as *CDKN2A* [10,11]. These additional mutations contribute to increased genomic instability, loss of differentiation markers, and enhanced proliferative and invasive capabilities.

While the molecular landscape of ATC overlaps significantly with that of DTC, the acquisition of secondary mutations and epigenetic modifications leads to the loss of thyroid-specific functions, such as iodine uptake, rendering traditional radioactive iodine therapy ineffective [6]. This highlights the importance of molecular profiling in understanding ATC pathogenesis and developing targeted therapeutic strategies.

Surgical intervention and radiotherapy have shown promise in improving survival, but effective tumor management requires a multimodal approach. Chemotherapy, typically involving taxanes in combination with platinum-based agents or anthracyclines, remains a treatment option, though resistance can develop. Targeted therapies, such as dabrafenib and trametinib, offer a promising approach for patients with *BRAF* and *MEK* mutations, helping to overcome chemotherapy resistance. Meanwhile, patients without these mutations may still have access to clinical trials exploring immunotherapy with anti-PD-1 and anti-PD-L1 agents [12]. Despite advances in available treatments, therapeutic options remain limited, underscoring the urgent need for novel early detection methods and innovative therapeutic strategies [13].

Understanding ATC’s molecular landscape is critical for developing effective therapies. Genetic mutations play a pivotal role in the tumor’s aggressive behavior, influencing its transformation, proliferation, and invasion [14]. Several key genetic alterations contribute to ATC pathogenesis and disease progression. While *BRAF* and *MEK* mutations are among the most well characterized and already serve as therapeutic targets, not all ATC patients harbor these mutations, highlighting the need for additional treatment strategies. Other commonly identified mutations in ATC include alterations in *TP53, PTEN, TERT, PIK3CA, EIF1AX, RAS, RET*, and the SWI/SNF complex.

We conducted a comprehensive literature search for this review using the PubMed database and Google to summarize studies investigating these mutations. We aim to provide an in-depth analysis of these genetic alterations, their role in thyroid cancer progression, and current therapeutic strategies targeting them. By expanding our understanding of ATC’s molecular profile, researchers can work toward personalized treatment approaches that improve patient outcomes and survival rates.

## 2. Methodology

We searched the PubMed database and Google search engine for relevant articles to include in this review, focusing on keywords related to the topic of interest, specifically the names of each genetic mutation—list mutations. We reviewed article titles and abstracts and removed any that did not pertain to the subject of interest. Full texts of relevant articles were examined to determine if they could be included in the literature review. Each genetic mutation is summarized with a brief description of its function, role in thyroid cancer—generally and specifically in ATC, and therapies and/or associations with other genetic mutations. Our methods are limited to using full-text articles from PubMed and Google searches. Clinical trials listed online from the National Cancer Institute (NCI) were also included for mutations that are under investigation or have already been approved as a target for therapy. 

## 3. Genetic Mutation Markers in ATC

Genetic mutations play a fundamental role in the disruption of normal cell function, leading to the development and progression of cancer. Identifying genetic mutations characteristic of ATC is key to early detection and effectively treating the disease. In this section, organized by functional pathway and summarized in Table 1, we explore several genetic mutations in ATC cells, highlighting their contribution to the tumor’s aggressive nature and therapeutic strategies to combat the disease.

### 3.1. MAPK/ERK Signaling Genes

#### 3.1.1. BRAF in Anaplastic Thyroid Cancer: Role, Detection, and Targeted Therapies

##### Overview of BRAF

BRAF (B-type Raf kinase) is a cytoplasmic serine-threonine protein kinase and the most potent activator of the MAPK signaling pathway among RAF kinases [16]. The activation of this pathway initiates a cascade of downstream events that regulate transcriptional reprogramming and cell growth [26]. The most common BRAF alteration is the *BRAFV600E* mutation, where valine is substituted with glutamic acid at a mutation-prone site. This change mimics the phosphorylation of residues T599 and S602, inducing a structural modification in the activation segment. Consequently, BRAF remains constitutively active, continuously phosphorylating downstream targets [16].

Mammalian cells express three RAF proteins: ARAF, BRAF, and CRAF (RAF1). While all play essential roles in normal cellular functions, BRAF is the most frequently mutated RAF kinase across various cancers. *BRAF* mutations occur in approximately 60% of melanomas, 60% of thyroid cancers, 15% of colorectal cancers, and 5–8% of non-small cell lung cancers [26].

##### Role of BRAF in Thyroid Cancer

The *BRAFV600E* mutation is highly specific to the classic variant of papillary thyroid carcinoma (PTC) and is absent in follicular and medullary thyroid cancers, as well as benign thyroid neoplasms. A meta-analysis of 29 studies, encompassing over 2000 thyroid cancer cases, reported an average *BRAF* mutation prevalence of 44% in PTC and 24% in ATC [16].

##### Detecting *BRAF* Mutations in ATC

Given its presence in various tumors, detecting *BRAFV600E* mutations is a valuable diagnostic tool in ATC. As with PTC, somatic *BRAF* mutations are frequently observed in ATC, occurring in approximately 40–70% of cases, depending on the study. Immunohistochemistry (IHC) for the *BRAFV600E* mutation—the most prevalent *BRAF* alteration—is specific and sensitive. Positive BRAF immunoreactivity can aid in the diagnosis of ATC; however, this mutation is not exclusive to ATC and is found in other tumor types as well. IHC may serve as a surrogate for molecular testing, particularly in the preoperative and early post-diagnostic periods, and has theranostic implications for guiding BRAF-targeted therapy. Nevertheless, IHC results do not perfectly correlate with mutation status, so confirmatory molecular testing is recommended [13].

According to the American Thyroid Association (ATA) guidelines, evaluation for the *BRAFV600E* mutation should be initiated promptly once ATC is suspected—first with IHC, followed by molecular testing for confirmation. Molecular testing should be performed at diagnosis to inform treatment decisions, especially considering the availability of FDA-approved targeted therapies for *BRAFV600E*-positive tumors [27].

Genotyping through circulating tumor DNA (ctDNA) via liquid biopsy is an emerging approach for assessing *BRAFV600E* mutations or fusions in ATC patients. This method offers valuable insights into mutational profiles at diagnosis and for monitoring responses to targeted therapies. Further research is necessary to develop reliable clinical-grade assays for routine tumor DNA genotyping from blood samples [13].

##### Targeted Therapies for *BRAF*-Mutated ATC

The molecular profiling of ATC is increasingly used to guide personalized treatment. A phase II clinical trial demonstrated a significant response in ATC patients harboring *BRAFV600E* mutations when treated with dabrafenib (a BRAF inhibitor) and trametinib (a MEK inhibitor). In May 2018, the U.S. FDA approved this combination therapy for *BRAFV600E*-mutated ATC, excluding patients with other genetic alterations. (NCT02034110). This targeted therapy has demonstrated efficacy in patients with metastatic or locally advanced, unresectable stage IVC ATC harboring the *BRAFV600E* mutation. It is now considered the standard of care and is often used as a neoadjuvant treatment before surgery, with reported two-year overall survival rates reaching 80% [13,28,29].

For patients with unresectable *BRAFV600E*-mutated stage IVB ATC, standard treatment involves upfront chemoradiation. However, in cases where chemoradiation is not feasible or preferred, BRAF-targeted therapy is a viable alternative. Additionally, the neoadjuvant use of dabrafenib and trametinib is under investigation to render initially unresectable tumors operable. The preliminary findings suggest that surgical resection following a positive response to neoadjuvant BRAF inhibition significantly improves survival, with one study (n = 20) reporting a 94% one-year overall survival rate [13].

##### Ongoing Clinical Trials for *BRAF*-Mutated ATC Patients 

A phase I clinical trial (NCT0397523) currently evaluates the combination of dabrafenib, trametinib, and intensity-modulated radiation therapy (IMRT) in *BRAF*-mutated ATC patients. This trial assesses whether combining targeted inhibitors with radiation can enhance tumor control and improve patient outcomes [30].

By integrating precision diagnostics, targeted therapies, and emerging treatment strategies, ongoing research continues to advance the management of ATC, offering new hope for patients with this aggressive disease.

#### 3.1.2. MEK in Anaplastic Thyroid Cancer: Role and Targeted Therapies

##### Overview of MEK

Mitogen-activated protein kinase (MEK) plays a crucial role in regulating cellular functions such as proliferation, differentiation, and development, primarily through activation of the ERK signaling cascade [31]. MEK inhibitors have shown potential in thyroid cancer treatment by enhancing iodine uptake and retention, inducing G0/G1 cell cycle arrest through reduced MEK/ERK phosphorylation, and inhibiting the viability of cells harboring *BRAF* mutations. Additionally, a negative regulator of glycolysis associated with thyroid cancer growth utilizes the RAF/MEK/ERK pathway to increase glycolysis via GLUT1 overexpression while simultaneously suppressing mitochondrial respiration in thyroid cells [32].

##### Role of MEK in Thyroid Cancer

Various oncogenic alterations, including ALK translocations and HER2/3 mutations, can drive tumor proliferation through MEK activation, a key component of the MAPK pathway. This has led to extensive research on the therapeutic potential of MEK inhibition, regardless of specific genetic mutations. Recently, *ALK* gene translocations were identified in patient-derived thyroid cancer cells. Unlike the diverse ALK fusion variants seen in other cancers, thyroid cancer cells specifically exhibit *STRN-ALK* fusions, which encode for striatin. These hybrid mutations occur in approximately 4% of anaplastic thyroid carcinomas (ATCs) and 9% of poorly differentiated thyroid cancers, resulting in sustained activation of the MAPK pathway through MEK signaling [17].

The *STRN-ALK* fusion is a genetic alteration implicated in thyroid cancer, leading to the downregulation of sodium-iodide symporter (NIS), a key factor in radioactive iodine (RAI) refractoriness. The loss of NIS expression reduces iodine uptake, rendering RAI therapy ineffective. However, studies suggest that ALK inhibitors can restore NIS expression, potentially reversing RAI refractoriness and improving therapeutic outcomes [33].

##### Role of MEK in ATC

The MAPK-MEK signaling pathway is frequently hyperactivated in ATC and strongly associated with disease progression [32]. Targeting MEK in ATC has gained significant interest due to its role in sustaining tumor growth and resistance mechanisms.

##### MEK-Targeted Therapies

Multiple MEK inhibitors are currently under investigation for their efficacy in advanced thyroid cancers. Since thyroid tumors can develop resistance to RAF inhibitors by relieving negative feedback mechanisms, combining RAF and MEK inhibitors offers a rational strategy to effectively target the MAPK pathway [34].

A clinical trial (NCT02034110) sponsored by Novartis Pharmaceuticals evaluated the combination of dabrafenib (a BRAF inhibitor) and trametinib (a MEK inhibitor) in 16 patients with *BRAFV600E*-mutant ATC. The overall response rate was 69%, demonstrating promising efficacy. However, the treatment also exhibited a mixed safety profile, with grade 3–4 toxicities affecting up to 19% of patients, though all adverse effects were manageable. Interestingly, in *BRAFV600E*-driven differentiated thyroid carcinoma, the same combination therapy yielded a lower response rate of 30%, which was unexpected given the aggressive nature of ATC [17,34].

Overcoming resistance in *BRAF*-mutated PTC and ATC remains challenging. MEK inhibitors (e.g., selumetinib, cobimetinib, and trametinib), used alone or with BRAF inhibitors, rarely prevent progression, as ERK1/2 reactivation often occurs via bypass signaling. New strategies target ERK1/2 directly; the ERK inhibitor ulixertinib (BVD-523) has shown promise in trials and received expanded access for *BRAF*-mutant melanoma. However, resistance mechanisms, including compensatory MEK5–ERK5 activation, have already been identified [35].

Notably, Iyer et al. demonstrated that ATC patients treated with lenvatinib or dabrafenib-trametinib experienced renewed responses when pembrolizumab, a PD-L1 inhibitor, was added after progression. This suggests that in select cases, immunotherapy may enhance the efficacy of kinase inhibitors [36].

##### Ongoing Clinical Trials for *MEK*-Mutated ATC Patients

In addition to clinical trials assessing dabrafenib and trametinib, researchers are exploring novel therapeutic strategies targeting MEK and related pathways. A phase II clinical trial (NCT060077924) is currently investigating the combination of avutometinib, a RAF/MEK clamp, and defactinib, a focal adhesion kinase (FAK) inhibitor [37]. These agents aim to disrupt kinase-driven tumor growth, potentially improving treatment outcomes for ATC patients [38].

As research continues to uncover the molecular drivers of ATC, targeting the MEK pathway through innovative combination therapies holds promise for enhancing treatment efficacy and overcoming resistance in this highly aggressive malignancy.

#### 3.1.3. *RAS* Mutations in Anaplastic Thyroid Cancer: Role and Detection Methods

##### *RAS* Family Mutations

*NRAS*, *HRAS*, and *KRAS* mutations play a crucial role in thyroid cancer pathogenesis [39]. These genes are located on chromosome 1 (*HRAS*), chromosome 12 (*KRAS*), and chromosome 11 (*NRAS*), and they encode proteins involved in key signaling pathways such as MAPK and PI3K-AKT, which regulate cell differentiation, proliferation, and survival [40].

##### Role of RAS in Thyroid Cancer

*RAS* mutations are particularly prevalent in follicular thyroid carcinoma (FTC), occurring in approximately 30–45% of cases [41]. They are less common in classical papillary thyroid carcinoma (PTC) but are detected in 20–40% of poorly differentiated thyroid cancers (PDTCs) and 10–20% of anaplastic thyroid cancers (ATCs) [18]. In well-differentiated thyroid cancers (WDTCs), *RAS* mutations are considered key molecular markers with diagnostic and prognostic significance [42]. While often associated with a more indolent disease course, *RAS* mutations and other alterations, such as *TERT* promoter mutations, can indicate a more aggressive clinical behavior and poorer prognosis [18].

##### Role of RAS in ATC

Although *RAS* mutations are less frequent in ATC than other subtypes, their presence suggests a role in tumor progression and aggressive disease behavior [43]. *RAS* mutations are also not currently central to the diagnosis of anaplastic thyroid carcinoma (ATC); however, their consideration may offer important insights—especially in cases with overlapping features of poorly differentiated thyroid carcinoma (PDTC) or when evaluating targeted therapy options.

In ATC, the prevalence of mutations includes *BRAF* (45%), *RAS* (24%), *TERT promoter* (75%), *TP53* (63%), *PIK3CA* (18%), *EIF1AX* (14%), and *PTEN* (14%). Notably, co-occurring *BRAF* or *RAS* and *TERT* promoter mutations are associated with significantly worse outcomes compared to single-gene alterations. Interestingly, *BRAF*- and *RAS*-mutated ATCs show similar rates of nodal and distant metastasis, suggesting comparable aggressiveness [9].

Emerging evidence indicates that *RAS* mutations, though more commonly associated with follicular thyroid carcinoma (FTC) and poorly differentiated thyroid carcinoma (PDTC), also play a role in the pathogenesis and progression of anaplastic thyroid carcinoma (ATC). Studies have shown that *RAS* mutations are present in approximately 23% of ATC cases, suggesting their involvement in the disease’s development [44].

Given the genetic complexity of ATC and findings that mutations in *BRAF, PTEN*, *PI3KCA, RAS,* and *TP53* are frequently observed, incorporating *RAS* status into the diagnostic and therapeutic framework may improve patient stratification and guide personalized treatment strategies within a multidisciplinary approach [45]

##### Detection of RAS Mutations in ATC

The clinical significance of *RAS* mutations in thyroid cancer includes their use in fine-needle aspiration biopsy (FNAB) to help distinguish between benign and malignant thyroid nodules, particularly in cases with indeterminate cytology [46]. *RAS* mutations in these nodules are often associated with a higher likelihood of malignancy, guiding more aggressive surgical management [47].

#### 3.1.4. *NTRK* Fusions in Anaplastic Thyroid Cancer: Role and Therapeutic Implications

##### Overview of the NTRK Family

Neurotrophic tropomyosin kinase receptors (NTRKs) are transmembrane tyrosine kinases important for the development of the nervous system and are encoded by the *NTRK1, NTRK2*, and *NTRK3* genes. Mutations in these genes, known as fusions, activate several intracellular pathways, including the MAPK pathway contributing to the aggressive nature of cancer. While not all *NTRK* fusions are associated with poor prognosis, they may contribute to a resistance to tyrosine kinase inhibitors used for therapy [48,49].

##### Role of NTRK in Thyroid Cancer

*NTRK* gene fusions are rare in thyroid cancer, occurring in approximately 2.3–3.4% of cases [50]. *NTRK* fusions involve the fusion of *NTRK1, NTRK2*, or *NTRK3* with various partner genes, leading to constitutive activation of the TRK signaling pathway, which promotes tumor growth, survival, and metastasis. In thyroid cancer particularly, *NTRK1* and *NTRK3* fusions are the most prevalent. *NTRK* fusions are found in both adult and pediatric PTC cases (1–6% and 4–26%, respectively) [51].

##### Role of NTRK in ATC

*NTRK* fusions have also been identified in anaplastic thyroid carcinoma (ATC). Xu et al. reported two patients with *NTRK* fusions (3%, n = 102), specifically *NTRK1-IRF2BP2* and *NTRK2-CRNDE* [9]. The presence of *NTRK* fusions in ATC suggests a potential role in tumor progression and dedifferentiation, making them a promising therapeutic target.

##### Therapeutic Implications of TRK Inhibitors for ATC

The recent development of TRK inhibitors, such as Larotrectinib and Entrectinib, has provided effective treatment options for *NTRK* fusion-positive cancers, offering new hope for patients with aggressive or refractory disease [52]. While there is no approved therapy for patients with ATC using TRK inhibitors, patients may respond well to this type of therapy, especially using Larotrectinib, which has an overall high response and tolerance [53].

### 3.2. PI3K/AKT Pathway Genes

#### 3.2.1. *PIK3CA* Mutations in Anaplastic Thyroid Cancer: Role and Targeted Therapies

##### Overview of PIK3CA

PIK3CA (Phosphatidylinositol-4,5-Bisphosphate 3-Kinase Catalytic Subunit Alpha) is a key component of the PI3K/AKT/mTOR signaling pathway, which is frequently dysregulated in thyroid cancer. Under normal physiological conditions, PIK3CA encodes the p110α catalytic subunit of phosphatidylinositol 3-kinase (PI3K), which is activated by growth factors binding to receptor tyrosine kinases. Once activated, PI3K phosphorylates PIP2 to generate PIP3, leading to the recruitment and activation of AKT and downstream effectors involved in cell growth, proliferation, survival, and migration [54].

##### Role of PIK3CA in Thyroid Cancer

PIK3CA activation in thyroid cancer can occur through multiple mechanisms, including activating mutations, gene amplification, and upstream activation by *RAS* mutations or *PTEN* loss [55]. These alterations drive thyroid cancer progression by promoting uncontrolled cell proliferation, enhanced survival, angiogenesis, metastasis, and dedifferentiation, particularly in aggressive tumor subtypes [54,55]. The frequency of *PIK3CA* mutations and amplifications varies across thyroid cancer subtypes, with higher rates observed in PDTC and ATC [19,54].

##### Role of PIK3CA in ATC

The relatively high prevalence of *PIK3CA* alterations in ATC (~23%) underscores the role of PI3K pathway activation in tumor dedifferentiation and progression to more aggressive phenotypes [19].

##### Therapeutic Implications of PI3K Inhibitors for ATC

Insights into the role of the PI3K/AKT pathway in thyroid cancer have led to the development of targeted therapies, including PI3K inhibitors. These inhibitors are currently being investigated for treating advanced thyroid cancers with *PIK3CA* alterations [56]. These inhibitors aim to disrupt tumor growth and survival mechanisms driven by PI3K pathway activation, offering potential therapeutic options for ATC patients.

#### 3.2.2. *PTEN* Mutations in Anaplastic Thyroid Cancer: Role and Associations

##### PTEN Function

Phosphatase and tensin homolog (PTEN) is a tumor suppressor that negatively regulates the PI3K/AKT pathway, controlling cell survival, proliferation, metabolism, and structure. *PTEN* loss, through somatic deletions or loss of heterozygosity (LOH), is common in various thyroid cancer subtypes [57].

##### Role of PTEN in Thyroid Cancer

In papillary thyroid cancer (PTC), *PTEN* loss contributes to tumor progression by reducing p-AKT levels and increasing angiogenesis [58]. Additionally, PTEN and PPARG inactivation is linked to heightened aggressiveness and NF-kB activation in thyroid cancer [59]. *PTEN* mutations are also associated with Cowden syndrome, where over 80% of patients inherit *PTEN* mutations, leading to thyroid cancer susceptibility [57].

##### Role of PTEN in ATC

Alterations in the PI3K/AKT pathway occur in 30–40% of ATCs, with *PTEN* mutations found in 10–15% [13]. The oncogenic miR-17-92 cluster promotes ATC progression by targeting *PTEN*, suppressing apoptosis [59]. *PTEN* loss often coexists with *RAS* and *NF1* mutations in ATC, while *PIK3CA* mutations frequently accompany *BRAFV600E* mutations [60].

##### Associations with PTEN

*PTEN* mutations are observed in approximately 15% ATCs and are typically associated with functional inactivation of the PTEN tumor suppressor. This loss of function contributes to constitutive activation of the PI3K/AKT signaling cascade. These alterations may arise through either germline or somatic events. Next-generation sequencing (NGS) studies show that *PIK3CA* and *AKT1* mutations dominate advanced disease stages, while *PTEN* alterations occur in FTCs and follicular adenomas [60]. In Cowden syndrome, *PTEN* inactivation elevates FTC risk through mutations, deletions, hypermethylation, or post-translational modifications, making PTEN a crucial predictive marker for thyroid cancer [57].

#### 3.2.3. *RET* Mutations in Anaplastic Thyroid Cancer: Role and Targeted Therapies

##### RET Function

Rearranged During Transfection (RET) is a receptor tyrosine kinase that regulates cell growth, differentiation, and survival, particularly in neural crest-derived tissues. While primarily membrane-bound, RET can translocate to the cytoplasm and nucleus upon activation [61]. Mutations in RET activate key oncogenic pathways, including PI3K/AKT, RAS/RAF/MAPK, and PLCγ, driving tumor proliferation and growth [62].

##### Role of RET in Thyroid Cancer

*RET* mutations are more prevalent in medullary thyroid cancer (MTC) than in papillary thyroid carcinoma (PTC). A European study reported that 75% of MTC patients were tested for *RET* mutations, while *RET* alterations in PTC occur in 10–25% of cases, with potentially higher rates among radiation-exposed individuals [61].

##### Role of RET in ATC

Although *RET* fusions are uncommon in anaplastic thyroid cancer (ATC), they are associated with aggressive tumor behavior, including increased rates of lymph node and distant metastases. A study by Xu et al. found *RET* mutations in 2% of ATC samples [9,63].

##### Ongoing Clinical Trials for *RET*-Mutated ATC Patients

A phase II clinical trial is currently evaluating the combination of lenvatinib, a multi-targeted tyrosine kinase inhibitor, and pembrolizumab, an immunotherapy agent, in patients with unresectable stage IVB ATC. Lenvatinib targets multiple pathways, including RET, to inhibit tumor growth, while pembrolizumab enhances immune response. This combination is being explored for its potential to improve treatment efficacy compared to monotherapy [64,65].

### 3.3. Cell Cycle Regulation Genes

#### 3.3.1. *Tumor Protein p53* (*TP53*) Mutations in Anaplastic Thyroid Cancer: Role and Therapeutic Implications

##### p53 Function

*Tumor protein p53 (TP53*) is often regarded as a key regulator of cellular processes. It encodes the p53 protein, commonly referred to as the “guardian of the genome” due to its essential role in preserving genomic stability. Beyond this, p53 also has a caretaker function, participating in various DNA repair mechanisms. However, mutant p53 can acquire oncogenic properties, promoting cancer progression through mechanisms independent of its normal tumor-suppressing functions. In thyroid cancer, *TP53* mutations are most frequently observed in exons 5–8. Elevated p53 protein levels have been associated with thyroid cancer, as immunohistochemical studies reveal increased expression in anaplastic, poorly differentiated, and well-differentiated thyroid tumors [57].

##### Role of p53 in Thyroid Cancer

A study examining a *BRAFV600E*-mutated ATC transplant model found that p53 expression increased more than fivefold compared to a two-month-old primary tumor. Additionally, p53 expression progressively rose from the early to late stages, with lower levels at two months and higher levels at four to six months. This pattern suggests a compensatory response by p53 to counteract tumor progression driven by BRAFV600E. These findings indicate that *BRAFV600E*-induced senescence plays a crucial role in tumor regression, mediated by p53 [66].

##### Role of p53 in ATC

*TP53* mutations are highly prevalent in ATC, occurring in 50–80% of cases. The inactivation of p53, either through mutations or other mechanisms, may contribute to the progression from well-differentiated thyroid cancer to ATC, suggesting that *TP53* alterations could represent a later event in cancer development [20,57]. To further investigate this, Zou et al. developed two mouse models: *TPO–BRAFV600E–Trp53*−/− (homozygous *Trp53* knockout) and TPO–*BRAFV600E–Trp53*+/− (heterozygous *Trp53* knockout). Both models developed ATC, but tumors in the TPO–*BRAFV600E–Trp53*+/− mice exhibited a 2–3-month delay in onset and slower growth compared to those in the TPO–*BRAF V600E*–*Trp53*−/− mice, with no loss of the wild-type p53 allele. ATC transformation was observed as early as 12 weeks, suggesting that the inactivation of a single *Trp53* allele is sufficient to drive ATC transformation and tumor growth independently of TSH [66].

##### Therapeutic Implications of Restoring p53 for ATC

Emerging therapeutic strategies for targeting tumor suppressor genes (TSGs) focus on modulating TSG activity to suppress oncogenic pathways and exploit the effects of TSG loss in cancer cells. Thyroid cancer represents an ideal candidate for gene therapy due to (i) the use of tumor-specific promoters to drive therapeutic gene expression selectively in cancer cells, reducing off-target effects, and (ii) the feasibility of comprehensive thyroid hormone replacement therapy for patients [57].

Restoring wild-type p53 function in thyroid cancer cells has been shown to reinstate critical cellular functions and counteract tumor progression [57]. For instance, introducing wild-type p53 into the anaplastic thyroid cancer-derived ARO cell line (which harbors mutated p53) significantly reduced cell proliferation and increased the proportion of cells in the G0/G1 phase. This intervention also diminished the tumorigenic potential and enhanced responsiveness to TSH, as evidenced by elevated levels of thyroglobulin, thyroid peroxidase, and TSH receptor expression [21].

However, a study by Fagin et al. demonstrated limited stable transfection success when introducing wild-type p53 into clonal undifferentiated thyroid carcinoma cell lines with mutated p53, with only one clone successfully expressing wild-type p53 and thyroid peroxidase. Despite this challenge, these findings underscore the pivotal role of p53 in maintaining thyroid tumor cell differentiation. Additionally, p53 has been implicated in activating immune responses contributing to tumor suppression [57,67].

Different p53 mutations contribute to immune suppression through multiple mechanisms. Loss-of-function mutations in *TP53* impair tumor cell apoptosis and promote an immunosuppressive tumor microenvironment (TME) by enhancing TGF-β signaling and regulatory T-cell (Treg) infiltration [68]. Gain-of-function (GOF) mutations can upregulate PD-L1, suppressing T-cell activity, and enhance secretion of immunosuppressive cytokines like IL-10 and TGF-β [69]. Additionally, mutant p53 enhances tumor-associated macrophage (TAM) polarization towards the M2 phenotype, further suppressing anti-tumor immunity [68]. These mechanisms collectively reduce anti-tumor immunity, facilitating cancer progression.

#### 3.3.2. *CDKN2A* Mutations in Anaplastic Thyroid Cancer: Role and Associations

##### Overview of CDKN2A

The *CDKN2A* (cyclin-dependent kinase inhibitor 2A) gene encodes multiple proteins, including the tumor suppressors p16^INK4A^ and p14^ARF^, which have distinct functions. These proteins arise from alternative splicing of the *CDKN2A* gene and are regulated by separate promoters [70].

In response to oncogenic signals, p14^ARF^ is crucial in supporting the tumor suppressor protein p53 by binding to and sequestering Mdm2 in the nucleolus. Since Mdm2 promotes p53 degradation, its sequestration by p14^ARF^ leads to increased p53 levels, thereby enhancing p53-mediated tumor suppression, including cell cycle arrest, senescence, and apoptosis [71].

Meanwhile, p16INK4A functions as a cyclin-dependent kinase inhibitor, blocking cyclin D-CDK4 and cyclin D-CDK6 complexes from phosphorylating the tumor suppressor protein pRb. In its non-phosphorylated or hypo-phosphorylated state, pRb inhibits transcription factors such as E2F, preventing the activation of genes required for cell cycle progression and restricting the transition past the G1/S phase checkpoint [72].

The *CDKN2A* gene is located at the 9p21.3 cytogenetic locus, along with *CDKN2B*, another tumor suppressor gene [73]. Immunohistochemical studies have shown that both p16^INK4A^ and p14^ARF^ are predominantly localized in the nucleus [70].

##### Role of CDKN2A in Thyroid Cancer

The activity of CDKN2A is often diminished or entirely lost in thyroid cancer due to mutations, homozygous deletions, and promoter hypermethylation [71]. A genetic analysis of 779 advanced differentiated and anaplastic thyroid cancers revealed CDKN2A inactivation in approximately 7% of advanced differentiated thyroid cancers [22]. Another study found that while *CDKN2A* alterations were present in some advanced differentiated thyroid cancers, they were absent in minimally invasive follicular and papillary thyroid cancer specimens [74].

##### Role of CDKN2A in ATC

Pozdeyev et al. reported that *CDKN2A* and *CDKN2B* alterations were significantly more common in ATC, occurring in 22% and 13% of cases, respectively. This makes *CDKN2A* the second most frequently altered tumor suppressor gene in ATC, following *TP53* [22,74]. The most prevalent alterations included loss-of-function mutations and deletions, with the 9p21.3 locus being ATC’s most frequently affected copy-loss region [74].

A genomic analysis of 101 ATC specimens by Xu et al. found *CDKN2A/CDKN2B* gene alterations in 25% of tumors (25/101), with 19 cases exhibiting deep deletions. The loss of *CDKN2A* copy number and the absence of p16^INK4A^ were significantly associated with reduced disease-specific survival in patients with ATC and advanced differentiated thyroid cancers [9]. Additionally, a transcriptome analysis revealed that *CDKN2A* deletions in ATC correlated with lower thyroid differentiation scores, suggesting that CDKN2A status may be a prognostic marker for advanced differentiated and anaplastic thyroid cancer [74].

##### Associations with *CDKN2A*

In ATC specimens with *CDKN2A* copy number loss, increased expression of the PD-L1 and PD-L2 encoding genes, *CD274* and *PDCD1LG2*, respectively, was observed [74]. Given that elevated PD-L1 protein levels are linked to immune evasion in tumor cells, targeting PD-L1 in ATC cases with *CDKN2A* deletions may represent a promising immunotherapeutic strategy.

### 3.4. Telomere Maintenance

#### 3.4.1. *TERT* Mutations in Anaplastic Thyroid Cancer: Role and Associations

##### Overview of TERT

Telomerase Reverse Transcriptase (TERT) is the catalytic subunit of telomerase, an enzyme responsible for maintaining telomere length by adding telomeric repeats to chromosome ends. This function prevents cellular senescence and supports cell immortality [24]. While TERT is primarily localized in the nucleus, it can also be found in the cytoplasm and mitochondria [75].

##### Role of TERT in Thyroid Cancer

*TERT* promoter mutations are detected in approximately 10% of thyroid cancer cases. Liu et al. reported the combined prevalence of C228T and C250T mutations as follows: 11.7% in papillary thyroid carcinoma (PTC) (30/257), 13.9% in follicular thyroid carcinoma (FTC) (11/79), 37.5% in poorly differentiated thyroid carcinoma (PDTC) (3/8), and 40.1% in anaplastic thyroid carcinoma (ATC) (25/54) [23].

##### Role of TERT in ATC

The prevalence of *TERT* promoter mutations increases progressively from well-differentiated thyroid cancers to ATC, with over 70% of ATCs harboring *TERT* promoter mutations [24]. These mutations frequently co-occur with *BRAFV600E* or *RAS* mutations, creating a synergistic effect that enhances tumor aggressiveness [76]. In differentiated thyroid cancers, the presence of *TERT* promoter mutations is linked to an increased risk of transformation to ATC [77]. Beyond its role in telomere maintenance, TERT also has non-canonical functions that contribute to cancer progression, including gene expression regulation and enhancement of the DNA damage response. In poorly differentiated and anaplastic thyroid cancers, *TERT promoter* mutations are often clonal events, underscoring their role in driving tumor evolution toward more aggressive phenotypes [78].

##### Associations with *TERT*

*TERT* promoter mutations significantly correlate with older age, larger tumor size, and male sex. In conventional PTC, they are also associated with lymph node metastasis and the *BRAFV600E* mutation [24].

### 3.5. Chromatin Remodeling

#### 3.5.1. *EIF1AX* Mutations in Anaplastic Thyroid Cancer: Role and Associations

##### Overview of EIF1AX

Eukaryotic Translation Initiation Factor 1A X-Linked (EIF1AX) is a crucial component of the translation initiation process. It plays a key role in scanning and selecting the AUG start codon, thereby influencing the translation of specific mRNAs. While primarily localized in the cytoplasm, it is also found in the nucleus. *EIF1AX* mutations are detected in approximately 14% of thyroid cancers, with reported frequencies of 11% in poorly differentiated thyroid cancer (PDTC) and 9% in anaplastic thyroid cancer (ATC) [25].

##### Role of EIF1AX in Thyroid Cancer

EIF1AX is essential for recruiting the ternary complex and assembling the 43S preinitiation complex (PIC) as part of the translation initiation complex. Mutations in *EIF1AX*, particularly the C-terminal EIF1AX-A113 splice variant, can stabilize the PIC and activate ATF4, a cellular stress sensor, which suppresses EIF2α phosphorylation and enhances overall protein synthesis [79].

##### Role of EIF1AX in ATC

*EIF1AX* mutations, particularly the A113 splice variant, contribute significantly to ATC by promoting protein synthesis, modifying cellular stress responses, and cooperating with other oncogenic drivers such as RAS. The A113 splice mutation enhances tumorigenesis through multiple mechanisms, including stabilization of the 43S preinitiation complex, upregulation of ATF4, and suppression of EIF2α phosphorylation, collectively leading to increased protein synthesis and tumor growth [79].

##### Association with *RAS*

*EIF1AX* mutations frequently co-occur with *RAS* mutations in advanced thyroid cancers, collectively driving tumorigenesis. The EIF1AX-A113 splice variant, in combination with RAS, stabilizes c-MYC, further accelerating tumor progression. ATF4 and c-MYC activation induced by *EIF1AX* mutations enhance amino acid transporter expression and increase mTOR sensitivity to amino acid availability, promoting tumor growth [79]. The presence of *EIF1AX* mutations, particularly alongside *RAS* alterations, is associated with a higher risk of malignancy and more aggressive tumor behavior [80].

#### 3.5.2. *SWI/SNF* Complex Mutations in Anaplastic Thyroid Cancer

##### SWI/SNF Complex Function

The *SWI/SNF* complex is a chromatin-remodeling complex that regulates gene expression by modifying chromatin structure [81].

##### Role of the SWI/SNF Complex in Thyroid Cancer

Loss of *SWI/SNF* complex function in thyroid cancer leads to reduced expression of thyroid differentiation genes, impaired radioiodine uptake, and the establishment of a repressive chromatin state that is unresponsive to MAPK pathway inhibition [22]. These mutations play a key role in radioiodine resistance in ATCs and contribute to the failure of MAPK inhibitor-based redifferentiation therapies. The presence of *SWI/SNF* mutations in ATCs underscores the importance of this complex in maintaining thyroid cell differentiation, and its loss confers resistance to conventional treatments, emphasizing the need for alternative therapeutic strategies [22,82].

##### Role of the SWI/SNF Complex in Anaplastic Thyroid Cancer (ATC)

A study analyzing 33 ATC samples found 36% harbored mutations in the *SWI/SNF* complex [25]. ATC exhibits alterations in multiple *SWI/SNF* subunit genes, including *ARID1A* and *ARID1B* (components of the BAF complex), *ARID2* (a component of the PBAF complex), and *SMARCB1* (shared between both BAF and PBAF complexes) [22,82].

## 4. Conclusions

Anaplastic thyroid cancer (ATC) is one of the most aggressive and deadly malignancies, marked by rapid progression, early metastasis, and resistance to standard therapies. With a median survival of only a few months after diagnosis, the prognosis remains grim, highlighting the urgent need for more effective treatment strategies.

This review places strong emphasis on integrating genomic data with clinical outcomes, focusing on how specific mutations can predict therapeutic responses and resistance patterns in ATC. It also uniquely connects mutation profiles with therapeutic decision-making. It goes beyond just identifying mutations and explores how understanding these alterations—especially in resistant ATC—can influence the choice of therapies, including combination therapies and immune checkpoint inhibitors. This approach adds value by offering a strategic framework for clinical decision-making in ATC management. There is a strong emphasis on integrating genomic data with clinical outcomes, focusing on how specific mutations (e.g., *BRAF, RAS, TP53*, and *TERT* promoter) can predict therapeutic responses and resistance patterns in ATC. This practical, clinically relevant perspective is underrepresented in the current literature.

This paper places strong emphasis on integrating genomic data with clinical outcomes, focusing on how specific mutations (e.g., *BRAF, RAS, TP53*, and *TERT* promoter) can predict therapeutic responses and resistance patterns in ATC. This practical, clinically relevant perspective is underrepresented in the current literature.

ATC’s genetic landscape is highly complex, with a substantial mutational burden contributing to its aggressive nature. Frequent genetic alterations include mutations in *BRAF*, *TP53*, *RAS*, the *TERT* promoter, and PI3K/AKT/mTOR pathway components. These mutations disrupt key cellular processes such as proliferation, apoptosis resistance, invasion, and immune evasion. Notably, co-occurring mutations—such as *BRAFV600E* and *TERT* promoter mutations—are associated with poorer clinical outcomes, suggesting that their synergistic oncogenic effects further accelerate disease progression.

Despite significant progress in understanding ATC at the molecular level, translating these insights into effective clinical treatments remains a major challenge. Conventional therapies, including surgery, radiation, and chemotherapy, often yield limited benefits due to the tumor’s aggressive nature and inherent resistance mechanisms. While targeted treatments, such as BRAF and MEK inhibitors, have shown promise in specific patient subgroups, their effectiveness is often short-lived as resistance inevitably develops. Similarly, immunotherapy, including immune checkpoint inhibitors, has produced mixed outcomes, likely due to ATC’s immunosuppressive tumor microenvironment.

The development of bypass variants is a well-known resistance mechanism to targeted therapy, where tumor cells activate alternative signaling pathways to evade treatment [83]. Combining immunotherapy with targeted therapy offers a synergistic approach by enhancing anti-tumor immunity while counteracting adaptive resistance. Immunotherapy can help eliminate resistant tumor clones by engaging cytotoxic T cells and reducing immunosuppressive signals in the tumor microenvironment [84]. Additionally, targeted therapy may enhance tumor immunogenicity by increasing neo-antigen presentation and disrupting immune evasion pathways, thereby improving the efficacy of immunotherapy [85]. This combinatorial strategy holds promises for more durable treatment responses and reduced relapse rates.

To improve patient outcomes, a multidisciplinary approach is essential—one that integrates molecular profiling, targeted therapies, immunotherapy, and advanced drug delivery systems. Personalized treatment strategies guided by comprehensive genomic and transcriptomic analyses can help identify actionable mutations and optimize therapeutic interventions. Additionally, novel combination therapies that target multiple oncogenic pathways and the tumor microenvironment hold potential for overcoming resistance and achieving more durable responses.

Ongoing research is critical to elucidate ATC’s molecular drivers further, enhance early detection, and develop innovative treatment approaches. By embracing precision medicine and fostering multidisciplinary collaboration, there is hope for more effective management—and ultimately, the eradication—of this devastating disease.

## Figures and Tables

**Table 1 jcm-14-02898-t001:** Summary of genetic mutations with mutation prevalence and associated therapies for ATC.

**Functional Pathway**	**Gene Mutation**	**Mutation Prevalence** **(By Study)**	**Associated Therapies for ATC** **(Approved/Experimental)**
MAPK/ERK Signaling	*BRAF*	24% (n = 94 [Total from 29 studies]); Xing [15], Ylli, Patel [16]	BRAF/MEK Inhibition with Dabrafenib & Trametinib (Phase II Clinical Trial [NCT02034110]; Approved)
			BRAF/MEK Inhibition with Dabrafenib & Trametinib and IMRT (Phase I Clinical Trial] [NCT03975231]; Experimental)
	*MEK*	4% (STRN-ALK Fusion mutations associated with MEK activation); Naoum, Morkos [17]	BRAF/MEK Inhibition with Dabrafenib & Trametinib (PHASE II Clinical Trial [NCT02034110]; Approved)
			BRAF/MEK Inhibition with Dabrafenib & Trametinib and IMRT (Phase I Clinical Trial] [NCT03975231]; Experimental)
			Avutometinib and Defactinib (Phase II Clinical Trial [NCT06007924]; Experimental)
	*RAS*	10–20%; Xing [18]	-
	*NTRK*	3% (n = 102; Xu, Fuchs [9])	-
PI3K/AKT Pathway	*PIK3CA*	23% (n = 70); García-Rostán, Costa [19]	PI3K Inhibitors (Experimental)
	*PTEN*	10–15%; Bible, Kebebew [13]	-
	*RET*	2% (n = 102); Xu, Fuchs [9]	Lenvatinib and Pembrolizumab for Stages IVB and IVC Anaplastic thyroid cancer (Phase II Clinical Trial [NCT04171622]; Experimental)
Cell Cycle Regulation	*TP53*	50–80%; Manzella, Stella [20]	Restoring wild-type p53 in human thyroid cancer cells (Experimental) Moretti, Farsetti [21]
	*CDKN2A*	22%; n = 196; Pozdeyev, Gay [22]	-
Telomere Maintenance	*TERT*	40.1% (n = 54); Liu, Bishop [23] 70%; Landa, Ganly [24]	-
Chromatin Remodeling	*EIF1AX*	9% (n = 33); Landa, Ibrahimpasic [25]	-
	*SWI/SNF* Complex	36% (n = 33); Landa, Ibrahimpasic [25]	-

## Data Availability

The data generated and analyzed in this study are available from the corresponding author upon reasonable request. Any publicly available datasets used in this research are cited within the manuscript.

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
