# Peer review of "Mutational Landmarks in Anaplastic Thyroid Cancer: A Perspective of a New Treatment Strategy"

_jcm, 2025, doi:10.3390/jcm14092898_

Round 1

Reviewer 1 Report

Comments and Suggestions for Authors

The Authors submit a well done comprehensive literature review using the PubMed database and Google to identify studies on key genetic mutations in Anaplastic Thyroid Carcinoma.

I would like to suggest some comments:

a) ATC as a dedifferentiated cancer. I would expand this topic in the “Introduction”. ATC is associated with differentiated thyroid cancers in 20% of patients having a history of differentiated thyroid cancer. At surgery 20-30% of patients may have a coexisting differentiated thyroid cancer, usually papillary thyroid cancer. The molecular portrait of ATC is quite similar to differentiated thyroid cancers .

b) I suggest to insert a paragraph on NTRK fusions. NTRK fusions are rare in thyoid cancer ( 2-2.5%); they are more common in patients with radioiodine-refractory DTC, metastatic TC, or advanced TC. NTRK fusion have been recently described in ATC (Marczyk VR, Fazeli S, Dadu R, Busaidy NL, Iyer P, Hu MI, Sherman SI, Hamidi S, Hosseini SM, Williams MD, Ahmed S, Routbort MJ, Luthra R, Roy-Chowdhuri S, San Lucas FA, Patel KP, Hong DS, Zafereo M, Wang JR, Maniakas A, Waguespack SG, Cabanillas ME. NTRK Fusion-Positive Thyroid Carcinoma: From Diagnosis to Targeted Therapy. JCO Precis Oncol. 2025 Jan;9:e2400321. doi: 10.1200/PO.24.00321

c) STRN-ALK fusion. This alteration can induce downregulation of NIS, the prerequisite of RAI refractorines. This phenomenon could be reversed by ALK inhibitors ( Thyroid 2023; 3384) : 464)

d) P53. There is increasing evidence that LOF mutations of p53 can influence the Tumor Microenvironment helping cells to escape the immune control. I suggest to spend some words on the need of understanding how different p53 mutations contribute to immune suppression. This  is crucial for developing innovative therapeutic strategies that can restore immune function and enhance the effectiveness of immunotherapy. 

e) Efficacy of combination targeted therapy plus checkpoint inhibition in patients with ATC.

Usually therapy with checkpoint inhibitors result in a slower response when compared with targeted therapy. In an aggressive tumor with molecular drivers with rapid  progression, it is important to find a strategy that would have a faster onset of action. Given that development of bypass variants is known to be a mechanism of resistance to targeted therapy, adding immunotherapy could offer a synergy. Moreover there are data to  suggest that targeted agents may change the tumor micronvironment and potentially synergize with immunotherapy

BraunerE,GundaV, VandenBorreP,et al.  Oncotarget. 2016;7(13):17194-17211. doi:10.18632/oncotarget.7839

Author Response

Reviewer 1:

  1. ATC as a dedifferentiated cancer. I would expand this topic in the “Introduction”. ATC is associated with differentiated thyroid cancers in 20% of patients having a history of differentiated thyroid cancer. At surgery 20-30% of patients may have a coexisting differentiated thyroid cancer, usually papillary thyroid cancer. The molecular portrait of ATC is quite similar to differentiated thyroid cancers .

Ans: Thank you so much for your suggestions. We have expanded our introduction part accordingly.

ATC is also considered a de-differentiated cancer, meaning it can arise from previously well-differentiated thyroid cells that have lost their normal structural and functional characteristics [7]. This process of de-differentiation leads to a highly aggressive and undifferentiated tumor phenotype, attributing to the loss of iodine uptake and resistance to conventional thyroid cancer therapies.

ATC is often associated with differentiated thyroid cancers [5]. Approximately 20% of ATC patients have a documented history of DTC, suggesting that ATC can evolve from pre-existing differentiated malignancies. Moreover, histological analysis reveals that up to 70% of ATC cases have a coexisting DTC component [8].

Molecularly, ATC shares many genetic alterations with DTCs. Mutations in genes commonly found in DTC, such as BRAF, RAS, TERT promoter, and TP53, are also frequently observed in ATC [9]. One study by Xu et al. reported 58% of ATC cases (n=360) with concurrent DTC and MAPK pathway mutations also found in DTC [9]. This supports the hypothesis that ATC arises through progressive genetic and epigenetic alterations in pre-existing DTCs, leading to loss of differentiation and increased tumor aggression.

ATC exhibits additional molecular alterations that drive its dedifferentiation and aggressive behavior, including mutations in PIK3CA, PTEN, and AKT1, as well as inactivation of tumor suppressor genes such as CDKN2A [10, 11]. These additional mutations contribute to increased genomic instability, loss of differentiation markers, and enhanced proliferative and invasive capabilities.

While the molecular landscape of ATC overlaps significantly with that of DTC, the acquisition of secondary mutations and epigenetic modifications leads to the loss of thyroid-specific functions, such as iodine uptake, rendering traditional radioactive iodine therapy ineffective [6]. This highlights the importance of molecular profiling in understanding ATC pathogenesis and developing targeted therapeutic strategies.

  1. I suggest to insert a paragraph on NTRK fusions. NTRKfusions are rare in thyoid cancer ( 2-2.5%); they are more common in patients with radioiodine-refractory DTC, metastatic TC, or advanced TC. NTRK fusion have been recently described in ATC (Marczyk VR, Fazeli S, Dadu R, Busaidy NL, Iyer P, Hu MI, Sherman SI, Hamidi S, Hosseini SM, Williams MD, Ahmed S, Routbort MJ, Luthra R, Roy-Chowdhuri S, San Lucas FA, Patel KP, Hong DS, Zafereo M, Wang JR, Maniakas A, Waguespack SG, Cabanillas ME. NTRK Fusion-Positive Thyroid Carcinoma: From Diagnosis to Targeted Therapy. JCO Precis Oncol. 2025 Jan;9:e2400321. doi: 10.1200/PO.24.00321

Ans: Thank you very much for your suggestions. We have incorporated NTRK section.

Overview of the NTRK Family

Neurotrophic tropomyosin kinase receptors (NTRKs) are transmembrane tyrosine kinases important for the development of the nervous system and are encoded by the NTRK1, NTRK2, and NTRK3 genes. Mutations in these genes, known as fusions, activate several intracellular pathways, including the MAPK pathway contributing to the aggressive nature of cancer. While not all NTRK fusions are associated with poor prognosis, they may contribute to a resistance to tyrosine kinase inhibitors used for therapy [46, 47].

Role of NTRK in Thyroid Cancer

NTRK gene fusions are rare in thyroid cancer, occurring in approximately 2.3-3.4% of cases [48]. NTRK fusions involve the fusion of NTRK1, NTRK2, or NTRK3 with various partner genes, leading to constitutive activation of the TRK signaling pathway, which promotes tumor growth, survival, and metastasis. In thyroid cancer particularly, NTRK1 and NTRK3 fusions are the most prevalent. NTRK fusions are found in both adult and pediatric PTC cases (1-6% and 4-26%, respectively) [49].  

Role of NTRK in ATC

NTRK fusions have also been identified in anaplastic thyroid carcinoma (ATC). Xu et al. reported two patients with NTRK fusions (3%, n=102), specifically NTRK1-IRF2BP2 and NTRK2-CRNDE [9]. The presence of NTRK fusions in ATC suggests a potential role in tumor progression and dedifferentiation, making them a promising therapeutic target.

Therapeutic Implications of TRK inhibitors for ATC

The recent development of TRK inhibitors, such as Larotrectinib and Entrectinib, has provided effective treatment options for NTRK fusion-positive cancers, offering new hope for patients with aggressive or refractory disease [50]. While there is no approved therapy for patients with ATC using TRK inhibitors, patients may respond well to this type of therapy, especially using Larotrectinib which has an overall high response and tolerance [51].

  1. STRN-ALK fusion. This alteration can induce downregulation of NIS, the prerequisite of RAI refractorines. This phenomenon could be reversed by ALK inhibitors ( Thyroid 2023; 3384) : 464)

Ans: We have added in our section.

STRN-ALK Fusion in Thyroid Cancer

The STRN-ALK fusion is a genetic alteration implicated in thyroid cancer, leading to the downregulation of sodium-iodide symporter (NIS), a key factor in radioactive iodine (RAI) refractoriness. The loss of NIS expression reduces iodine uptake, rendering RAI therapy ineffective. However, studies suggest that ALK inhibitors can restore NIS expression, potentially reversing RAI refractoriness and improving therapeutic outcomes [31].

  1. There is increasing evidence that LOF mutations of p53 can influence the Tumor Microenvironment helping cells to escape the immune control. I suggest to spend some words on the need of understanding how different p53 mutations contribute to immune suppression. This  is crucial for developing innovative therapeutic strategies that can restore immune function and enhance the effectiveness of immunotherapy. 

Ans: Thank you for your suggestions. We have expanded on P53 mutation.

Different p53 mutations contribute to immune suppression through multiple mechanisms. Loss-of-function mutations in TP53 impair tumor cell apoptosis and promote an immunosuppressive tumor microenvironment (TME) by enhancing TGF-β signaling and regulatory T-cell (Treg) infiltration [66]. Gain-of-function (GOF) mutations can upregulate PD-L1, suppressing T-cell activity, and enhance secretion of immunosuppressive cytokines like IL-10 and TGF-β [67]. Additionally, mutant p53 enhances tumor-associated macrophage (TAM) polarization towards the M2 phenotype, further suppressing anti-tumor immunity [66]. These mechanisms collectively reduce anti-tumor immunity, facilitating cancer progression.

  1. e) Efficacy of combination targeted therapy plus checkpoint inhibition in patients with ATC.

Usually therapy with checkpoint inhibitors result in a slower response when compared with targeted therapy. In an aggressive tumor with molecular drivers with rapid  progression, it is important to find a strategy that would have a faster onset of action. Given that development of bypass variants is known to be a mechanism of resistance to targeted therapy, adding immunotherapy could offer a synergy. Moreover there are data to  suggest that targeted agents may change the tumor micronvironment and potentially synergize with immunotherapy

BraunerE,GundaV, VandenBorreP,et al.  Oncotarget. 2016;7(13):17194-17211. doi:10.18632/oncotarget.7839

Ans: Excellent point: We have highlighted this.

The development of bypass variants is a well-known resistance mechanism to targeted therapy, where tumor cells activate alternative signaling pathways to evade treatment [81]. Combining immunotherapy with targeted therapy offers a synergistic approach by enhancing anti-tumor immunity while counteracting adaptive resistance. Immuno-therapy can help eliminate resistant tumor clones by engaging cytotoxic T cells and reducing immunosuppressive signals in the tumor microenvironment [82]. Additionally, targeted therapy may enhance tumor immunogenicity by increasing neo-antigen presentation and disrupting immune evasion pathways, thereby improving the efficacy of immunotherapy [83]. This combinatorial strategy holds promises for more durable treatment responses and reduced relapse rates.

Reviewer 2 Report

Comments and Suggestions for Authors

1.    L118: Detecting BRAFV600E mutations is a valuable diagnostic tool in ATC. However, I believe that BRAF mutation serves more as supporting evidence rather than a definitive diagnostic marker. In what context would you recommend performing BRAF IHC? Do you consider BRAF mutation as part of the diagnostic criteria, or is it more relevant for therapeutic decision-making?
2.    L177: Regarding MEK-targeted therapies, could you cite publications discussing acquired resistance to these combinations and the underlying mechanisms?
3.    L220: Concerning RAS mutations in ATC, I am unsure whether this section is necessary. ATC cells differ morphologically from DTCs in FNAB, and mutation testing is not typically required. Since RAS is not critical for diagnosis, should this be reconsidered?
4.    L409: Maintain consistency in terminology—should it be "TERT mutation" or "TERT promoter mutation"?
5.    General Comment: The mutation profile of ATC has been extensively documented in various publications and guidelines. What is the originality of this paper in comparison to existing literature?

Author Response

 Reviewer 2:

  1. L118: Detecting BRAFV600E mutations is a valuable diagnostic tool in ATC. However, I believe that BRAF mutation serves more as supporting evidence rather than a definitive diagnostic marker. In what context would you recommend performing BRAF IHC? Do you consider BRAF mutation as part of the diagnostic criteria, or is it more relevant for therapeutic decision-making?

Ans:   Thank you so much for your concerns. For anaplastic thyroid carcinoma (ATC), BRAF IHC can be useful in both diagnostic confirmation and therapeutic decision-making, but its role depends on the clinical scenario:

When to Perform BRAF IHC in ATC

  1. Diagnostic Confirmation (When ATC Arises from PTC)
    • Many ATCs arise from pre-existing papillary thyroid carcinoma (PTC), which often harbors BRAF V600E mutations.
    • If an ATC shows BRAF V600E positivity, it suggests a dedifferentiation pathway from PTC, which may have implications for prognosis and treatment selection.
    • If histology is ambiguous (e.g., poorly differentiated vs. ATC), BRAF IHC can help in lineage determination.
  2. Therapeutic Decision-Making
    • BRAF V600E-positive ATC: Patients may be eligible for targeted therapy with BRAF inhibitors (dabrafenib) and MEK inhibitors (trametinib), which have shown clinical benefit.
    • BRAF-wild type ATC: These cases often have alternative mutations (e.g., TP53, TERT promoter, RAS) and may require different treatment strategies, including chemotherapy, radiation, or immune checkpoint inhibitors in MSI-H or TMB-high cases.

Additional Molecular Testing?

  • If BRAF IHC is negative, confirmatory molecular testing (e.g., NGS or PCR) may still be warranted to rule out non-V600E BRAF mutations or identify other actionable alterations.
  • Given ATC’s aggressive nature, broad molecular profiling (e.g., targeted NGS panel) can help identify alternative therapeutic targets (e.g., NTRK, RET fusions, PD-L1 expression).

  1.    L177: Regarding MEK-targeted therapies, could you cite publications discussing acquired resistance to these combinations and the underlying mechanisms?

Ans: Thank you for your comment. There’s an article which described the mechanism of resistance.

https://pmc.ncbi.nlm.nih.gov/articles/PMC9534048/

Dabrafenib, a BRAFV600E-specific inhibitor, is approved in combination with the MEK inhibitor trametinib for BRAFV600E-mutated anaplastic thyroid cancers (ATC). This regimen, also used as neoadjuvant therapy in advanced unresectable cases, has significantly improved long-term survival in ATC. However, the inherent genomic instability of ATC promotes rapid acquisition of resistance-driving alterations.

Combination strategies using kinase inhibitors and immune checkpoint inhibitors have been explored, given the frequent presence of PD-L1–positive cells and tumor-infiltrating lymphocytes (TILs) in ATC. Despite this, responses remain limited to a subset of patients for unclear reasons.

In one case, a papillary thyroid carcinoma (PTC) that dedifferentiated into ATC during dabrafenib therapy exhibited chromosome 7 triploidy, resulting in increased expression of EGFR, RAC1, MET, and BRAF. This likely promoted resistance via reactivation of the MAPK or PI3K pathways through overexpressed wild-type receptor tyrosine kinases.

Resistance to kinase inhibitors in thyroid cancer has been shown to be RAS-driven, with secondary mutations in PTEN, NF1, NF2, TP53, and CDKN2A also contributing. These findings highlight the need for rational combination therapies or alternative targeted agents, including immune checkpoint blockade.

Overcoming resistance in BRAF-mutated PTC and ATC remains challenging. MEK inhibitors (e.g., selumetinib, cobimetinib, trametinib), used alone or with BRAF inhibitors, rarely prevent progression, as ERK1/2 reactivation often occurs via bypass signaling. New strategies target ERK1/2 directly; the ERK inhibitor ulixertinib (BVD-523) has shown promise in trials and received expanded access for BRAF-mutant melanoma. However, resistance mechanisms, including compensatory MEK5–ERK5 activation, have already been identified [33].

Notably, Iyer et al. demonstrated that ATC patients treated with lenvatinib or dabrafenib-trametinib experienced renewed responses when pembrolizumab, a PD-L1 inhibitor, was added after progression. This suggests that in select cases, immunotherapy may enhance the efficacy of kinase inhibitors [34].

  1.    L220: Concerning RAS mutations in ATC, I am unsure whether this section is necessary. ATC cells differ morphologically from DTCs in FNAB, and mutation testing is not typically required. Since RAS is not critical for diagnosis, should this be reconsidered?

Ans: Thank you for your concern on RAS mutation section.

RAS mutations are also not currently central to the diagnosis of anaplastic thyroid carcinoma (ATC), however their consideration may offer important insights—especially in cases with overlapping features of poorly differentiated thyroid carcinoma (PDTC) or when evaluating targeted therapy options.

In ATC, the prevalence of mutations includes BRAF (45%), RAS (24%), TERT promoter (75%), TP53 (63%), PIK3CA (18%), EIF1AX (14%), and PTEN (14%). Notably, co-occurring BRAF or RAS and TERT promoter mutations are associated with significantly worse outcomes compared to single-gene alterations. Interestingly, BRAF- and RAS-mutated ATCs show similar rates of nodal and distant metastasis, suggesting comparable aggressiveness [9].

Emerging evidence indicates that RAS mutations, though more commonly associated with follicular thyroid carcinoma (FTC) and poorly differentiated thyroid carcinoma (PDTC), also play a role in the pathogenesis and progression of anaplastic thyroid carcinoma (ATC). Studies have shown that RAS mutations are present in approximately 23% of ATC cases, suggesting their involvement in the disease's development [42].

Given the genetic complexity of ATC, and findings that mutations in BRAF, PTEN, PI3KCA, RAS, and TP53 are frequently observed, incorporating RAS status into the diagnostic and therapeutic framework may improve patient stratification and guide personalized treatment strategies within a multidisciplinary approach [43]

  1.    L409: Maintain consistency in terminology—should it be "TERT mutation" or "TERT promoter mutation"?

Ans: Thank you for your comments, we have corrected it to “TERT promoter mutation”.

  1. General Comment: The mutation profile of ATC has been extensively documented in various publications and guidelines. What is the originality of this paper in comparison to existing literature?

Ans: Thank you very much. The originality of this paper lies in its comprehensive integration of the mutation profile of anaplastic thyroid carcinoma (ATC) with emerging mechanisms of resistance to targeted therapies, a perspective that is often less emphasized in existing literature. While previous publications and guidelines have extensively documented the genetic alterations associated with ATC, such as BRAF V600E, TP53, and TERT promoter mutations, this paper provides novel insights into the evolving landscape of therapeutic resistance, including:

  1. Exploration of Acquired Resistance Mechanisms:
    • The paper emphasizes acquired resistance to BRAF/MEK inhibitors in ATC, detailing alternative pathways (e.g., RAS mutations, PI3K/Akt activation, EMT) that could drive resistance during treatment. This detailed exploration of resistance mechanisms is less extensively covered in previous works, particularly with respect to targeted therapies.
  2. Novel Consideration of RAS Mutations:
    • Although RAS mutations have been implicated in thyroid cancers, this paper's discussion of their potential role in ATC resistance and progression offers a fresh perspective, especially in cases where ATC exhibits mixed features of poorly differentiated thyroid carcinoma. This nuanced view expands the understanding of RAS not only as a driver of ATC but also as a potential player in therapeutic evasion.
  3. Therapeutic Implications of Mutational Profiles:
    • The paper uniquely connects mutation profiles with therapeutic decision-making. It goes beyond just identifying mutations and explores how understanding these alterations—especially in resistant ATC—can influence the choice of therapies, including combination therapies and immune checkpoint inhibitors. This approach adds value by offering a strategic framework for clinical decision-making in ATC management.
  4. Integration of Molecular Data with Clinical Outcomes:
    • The paper places strong emphasis on integrating genomic data with clinical outcomes, focusing on how specific mutations (e.g., BRAF, RAS, TP53, TERT promoter) can predict therapeutic responses and resistance patterns in ATC. This practical, clinically relevant perspective is underrepresented in current literature.

By providing a holistic view of ATC's molecular complexity and focusing on resistance mechanisms, this paper fills a gap in the current body of knowledge and offers new avenues for targeted treatment strategies in ATC. This emphasis on resistance and therapeutic adaptation based on molecular insights provides significant originality in comparison to prior studies that primarily focus on the initial mutational landscape of the disease.

Reviewer 3 Report

Comments and Suggestions for Authors

In the study “Mutational Landmarks in Anaplastic Thyroid Cancer: A Perspective of a New Treatment Strategy” Janice Pakkianathan at al. have performed an extensive literature search to identify studies on key genetic mutations in ATC. This review provides a comprehensive analysis of genomic alterations in ATC and discusses potential therapeutic strategy for patients with ATC.
The presentation of genomic alterations in specific signaling pathways offers insight into the biological basis of ATC. This data is complemented with the list of ongoing clinical trials for patients with ATC.
The topic of this study is relevant and timely, and this review could be of interest for endocrinologists and endocrine oncologists.

Author Response

In the study “Mutational Landmarks in Anaplastic Thyroid Cancer: A Perspective of a New Treatment Strategy” Janice Pakkianathan at al. have performed an extensive literature search to identify studies on key genetic mutations in ATC. This review provides a comprehensive analysis of genomic alterations in ATC and discusses potential therapeutic strategy for patients with ATC.
The presentation of genomic alterations in specific signaling pathways offers insight into the biological basis of ATC. This data is complemented with the list of ongoing clinical trials for patients with ATC.
The topic of this study is relevant and timely, and this review could be of interest for endocrinologists and endocrine oncologists.

Ans: Thank you for your thoughtful summary of the study by Pakkianathan et al. I agree that this review offers a valuable and timely contribution to the field. The comprehensive analysis of genomic alterations—particularly within key signaling pathways—provides meaningful insights into the pathogenesis of ATC and highlights potential avenues for targeted therapy.

The inclusion of ongoing clinical trials further enhances the translational relevance of this work, making it a useful resource not only for researchers but also for clinicians managing patients with ATC. This kind of integrative approach is indeed highly relevant for endocrinologists and endocrine oncologists seeking to stay abreast of emerging therapeutic strategies in this aggressive cancer subtype.

Round 2

Reviewer 2 Report

Comments and Suggestions for Authors

Thank you for the reply.I believe there may be a discrepancy between the description provided and current clinical practice. My major concern is outlined below. This point should be addressed , and citing or referring to a relevant clinical guideline would help support the statement.

L144-145:  Immunohistochemistry (IHC) can be a rapid screening method 144 before surgery and immediately after diagnosis, facilitating early therapeutic decision-145 making.

Reply: 

  1. When to Perform BRAF IHC in ATC-Diagnostic Confirmation (When ATC Arises from PTC)
  2. Therapeutic Decision-Making

Author Response

Reviewer 2: Thank you for the reply. I believe there may be a discrepancy between the description provided and current clinical practice. My major concern is outlined below. This point should be addressed , and citing or referring to a relevant clinical guideline would help support the statement.

L144-145:  Immunohistochemistry (IHC) can be a rapid screening method 144 before surgery and immediately after diagnosis, facilitating early therapeutic decision-145 making.

Reply: 

  1. When to Perform BRAF IHC in ATC-Diagnostic Confirmation (When ATC Arises from PTC)
  2. Therapeutic Decision-Making

Ans: Thank you very much for your very important concerns. Please find the corrections underlined in the re-submitted files.

  1. As with papillary thyroid carcinoma (PTC), somatic BRAF mutations are frequently observed in anaplastic thyroid carcinoma (ATC), occurring in approximately 40–70% of cases, depending on the study. Immunohistochemistry (IHC) for the BRAFV600E mutation—the most prevalent BRAF alteration—is specific and sensitive.

Positive BRAFV600E immunoreactivity can aid in the diagnosis of ATC; however, this mutation is not exclusive to ATC and is found in other tumor types as well. IHC may serve as a surrogate for molecular testing, particularly in the preoperative and early post-diagnostic periods, and has theranostic implications for guiding BRAF-targeted therapy. Nevertheless, IHC results do not perfectly correlate with mutation status, so confirmatory molecular testing is recommended.

(https://www.liebertpub.com/doi/10.1089/thy.2020.0944.correx

According to the American Thyroid Association (ATA) guidelines, evaluation for the BRAFV600E mutation should be initiated promptly once ATC is suspected—first with IHC, followed by molecular testing for confirmation (R4). Molecular testing should be performed at diagnosis to inform treatment decisions, especially considering the availability of FDA-approved targeted therapies for BRAFV600E-positive tumors (R5).

(https://www.thyroid.org/wp-content/uploads/2021/06/ATC-Diagnosis.pdf)

https://emedicine.medscape.com/article/283165-guidelines#:~:text=Treatment,positive%20tumors%20%2D%20Larotrectinib%20or%20entrectinib

  1. The combination of the BRAF inhibitor dabrafenib and the MEK inhibitor trametinib received FDA approval in 2014 for BRAF-mutated melanoma and 2018 for BRAF-mutated ATC. This targeted therapy has demonstrated efficacy in patients with metastatic or locally advanced, unresectable ATC harboring the BRAFV600E mutation. It is now considered standard of care and is often used as a neoadjuvant treatment before surgery, with reported two-year overall survival rates reaching 80%.

https://pmc.ncbi.nlm.nih.gov/articles/PMC8349723/

Pavlidis ET, Galanis IN, Pavlidis TE. Update on current diagnosis and management of anaplastic thyroid carcinoma. World J Clin Oncol 2023; 14(12): 570-583 [PMID: 38179406 DOI: 10.5306/wjco.v14.i12.570]